# Emerging Role of LY6E in Virus–Host Interactions

**DOI:** 10.3390/v11111020

**Published:** 2019-11-03

**Authors:** Jingyou Yu, Shan-Lu Liu

**Affiliations:** 1Center for Retrovirus Research, The Ohio State University, Columbus, OH 43210, USA; yu.2123@osu.edu; 2Department of Veterinary Biosciences, The Ohio State University, Columbus, OH 43210, USA; 3Department of Microbial Infection and Immunity, The Ohio State University, Columbus, OH 43210, USA; 4Viruses and Emerging Pathogens Program, Infectious Diseases Institute, The Ohio State University, Columbus, OH 43210, USA

**Keywords:** Ly6/uPAR, viruses, LY6E, virus–host interactions

## Abstract

As a canonical lymphocyte antigen-6/urokinase-type plasminogen activator receptor Ly6/uPAR family protein, lymphocyte antigen 6 complex, locus E (LY6E), plays important roles in immunological regulation, T cell physiology, and oncogenesis. Emerging evidence indicates that LY6E is also involved in the modulation of viral infection. Consequently, viral infection and associated pathogenesis have been associated with altered LY6E gene expression. The interaction between viruses and the host immune system has offered insights into the biology of LY6E. In this review, we summarize the current knowledge of LY6E in the context of viral infection, particularly viral entry.

## 1. Introduction

LY6E, also known as Thymic Shared Antigen-1 (TSA-1) or Stem Cell Antigen-2 (SCA-2), is a glycosylphosphatidyl-inositol (GPI)-anchored cell surface protein that is 133 amino acids in length. LY6E was initially identified in mouse thymus, where its expression was observed in a phenotypically immature or nonmature subpopulation of CD4^-^CD8^-^ thymocytes, implying that it may serve as a thymocyte marker to discriminate immature from mature thymocytes subsets [1]. The murine LY6E cDNA was subsequently cloned by two independent groups simultaneously [2,3], while the human counterpart was cloned shortly after, with a predicted structure highly resembling the mouse homologs and other Ly6/uPAR superfamily members [4,5]. LY6E is transcriptionally active in a number of tissues, including the liver, spleen, uterus, ovary, lung, and brain [5], and its expression can be induced by type I interferon (IFN). The primary function of LY6E has been associated with immune regulation, specifically in modulating T cell activation, proliferation, development, tumor metastasis, and differentiation [6,7,8,9]. Linking LY6E to viral infection did not occur until the early 2000s, and recently there has been increasing interest in the study for the role of LY6E in viral interactions. 

## 2. Genetic Association of LY6E with MDV and MAV-1 Infection

In 2001, LY6E was reported to be differentially expressed in Marek’s disease (MD) caused by an oncogenic avian herpesvirus, named Marek's disease virus (MDV) [10], in chicken embryo fibroblasts [11]. An MD-susceptible chicken line, which supports higher MDV titers, showed a significantly higher level of LY6E expression compared to a resistant line where MDV viremia levels were lower [10]. A follow-up study unraveled that LY6E is among the candidate genes that determine chicken susceptibility to MDV [12], showing that LY6E directly interacts with MDV pathogenic US10 protein [12]. However, no direct evidence was available regarding how this interaction might affect cellular physiology and viral pathogenesis [12]. It also remains elusive as to whether this specific US10–LY6E interaction is a true determinant for MD pathogenesis. Therefore, it will be informative to examine whether viruses closely related to MDV strains also have similar sensitivities to LY6E modulation.

The LY6E protein has also been linked to enhanced infection by mouse adenovirus type 1 (MAV-1). The susceptibility of mouse cells to MAV-1 infection was first linked to a locus Msq1 in mouse chromosome 15 [13]. Subsequent studies refined the target region in a 0.75 Mb interval that harbors 15 candidate genes, with eight encoding Ly6 family proteins (*Ly6e, Ly6i, Ly6a, Ly6c1, Ly6c2, Ly6g, Ly6f*, and *Ly6h*) and six encoding predicted proteins that have the LU domains *(gpihbp1, 2010109I03Rik-201, I830127L07Rik, BC025446, AC116498.15,* and *9030619P08Rik*); one or multiple members from Ly6 gene family may aid in governing the MAV-1 susceptibility [14,15]. However, further functional studies are needed to validate exactly which genes are responsible for the MAV-1 susceptibility.

## 3. Opposing Roles of LY6E in HIV Entry

Susceptibility to HIV-1 infection has been linked to a region in human chromosome where LY6/uPAR family proteins reside [16]. Loeuillet et al. utilized in vitro whole-genome analysis and refined an SNP rs2572886 on human chromosome 8q24 that contributes to high cellular susceptibility to HIV-1 infection in primary T cells. They identified eight highly responsible genes and tested each of them in vitro by using siRNA and HeLa-TZM-bl (generated from a HeLa cell line by introducing the luciferase and β-galactosidase genes under control of the HIV-1 promoter) indicator infection assays [16], but interestingly, they observed only a modest effect. It was of note that, in this study, the endogenous level of the LY6 family protein expression was not assessed. Moreover, highly permissive and physiologically non-relevant HeLa-TZM-bl cells may not recapitulate the natural HIV-1 infection in CD4^+^ T cells. 

A series of new studies have focused on the role of LY6E in HIV infection in more physiological settings. One study showed that LY6E expression in monocytes down-regulates CD14 and thus dampens the TLR4/CD14-dependent proinflammatory responses [17]. They found that the CD14 level in monocytes was lower in antiretroviral-naive subjects with a low CD4 count than in those with high CD4 counts, and that CD14 levels were partially restored in drug-treated individuals, indicating that CD14 expression is inversely correlated with LY6E in primary monocytes of subjects chronically infected with HIV [17]. However, no direct interaction between LY6E and HIV has been demonstrated, although some data seem to support the notion that LY6E is actively engaged in HIV-1 pathogenesis. We recently explored the direct role of LY6E in HIV-1 infection, particularly at the early stage of viral replication [18]. In primary human PBMCs, CD4^+^ T cells, as well as monocytic THP1- cells, we observed that LY6E promotes HIV-1 entry, likely through an enhanced virus–cell fusion process. While the exact mechanism remains to be elucidated, this enhancing effect of LY6E on HIV-1 entry appears to be associated with the lipid raft localization of LY6E ascribed to its GPI anchor. Because HIV-1 entry requires CD4 and coreceptors, both of which are also localized in lipid rafts [19,20,21], it is possible that LY6E may modulate the properties of membrane lipids thus affecting HIV entry. Indeed, by using specific pharmaceutical inhibitors, we were able to demonstrate that the expansion of viral fusion pore induced by HIV-1 Env is enhanced by LY6E [18]. The positive role of LY6E in promoting HIV fusion is supported by recent work showing that LY6E acts as a receptor for the mouse endogenous retroviral envelope Syncytin-A, an essential molecule that is involved in placentogenesis and embryo survival [22]. In this study, it was shown that the depletion of LY6E impairs the syncytiotrophoblast fusion and placental morphogenesis, leading to embryonic lethality in mice [23]. LY6E has also been shown to directly interact with syncytin-A, and a soluble recombinant form of LY6E blocks the syncytin-A-mediated cell–cell fusion [22]. Overall, these recent data strongly implicate a role of LY6E in enhancing viral fusion and entry into host cells. 

Somewhat surprisingly, we recently uncovered a new yet distinct effect of LY6E on HIV-1 infection in low CD4-expressing T cells (Figure 1). In Jurkat T cells and primary monocyte-derived macrophages (MDMs), where CD4 expression levels are low, we found that HIV-1 entry was inhibited by LY6E [24]. This negative phenotype of LY6E in low CD4 cells is contrary to what we have observed in high CD4-expressing cells, including PBMCs [18]. Further experiments revealed that the differential phenotype of LY6E on HIV-1 infection is dependent on the level of CD4 in target cells. When the level of CD4 on the cell surface is low or limited, such as in the case of monocyte-derived macrophages (MDMs), the ability of LY6E to down-regulate CD4 is predominant, leading to reduced virus binding therefore entry. Mechanistically, we found that LY6E is enriched in lipid rafts where it mobilizes the CD4 molecules into a non-raft microdomain, in addition to enhancing the CD4 endocytosis, which collectively contributes to the downregulation of CD4 from the plasma membrane. Overall, the new work revealed an interesting model where LY6E can function distinctly in HIV-1 infected cells: on the one hand, it promotes HIV infection in high CD4 cells, but on the other, it inhibits HIV infection when the CD4 level in target cells is low. The opposing effect of LY6E on HIV infection may have implications for understanding the role of LY6E in the early stage of HIV transmission in monocytes/MDMs/DCs, where CD4 expression is low, in contrast to the late stage of AIDS pathogenesis where the virus predominantly infects high-CD4 T cells (Figure 1).

## 4. Modulation of Other Viral Infections by LY6E: Yellow Fever Virus (YFV), Dengue Virus (DENV), Influenza A Virus (IAV), and Vesicular Stomatitis Virus (VSV)

In addition to HIV-1, LY6E has been associated with enhanced infection for a number of enveloped RNA viruses (Table 1). In an early ISG (IFN-stimulated gene)-overexpression screening, Schoggins et al. observed that infection by Yellow Fever Virus (YFV) and Dengue Virus (DENV) is enhanced in STAT1^-/-^ fibroblasts, but to a less extent in Huh7.5 cells; this again emphasizes the cell type-dependent phenotype of LY6E as discussed above for HIV-1 [25,26]. In this study, the authors observed increased percentages of YFV-positive cell populations but not the level of YFV expression in individual infected cells, implying that LY6E likely influences an early stage of the viral infection. A follow-up study expanded the spectrum of viruses tested, and they found that only certain types of viruses are sensitive to the enhancement by LY6E. For example, infection by DENV, Zika Virus (ZIKV), Influenza A Virus (IAV), and Vesicular Stomatitis Virus (VSV) were elevated in LY6E-overexpressing STAT1^-/-^ fibroblasts, whereas infection by Sindbis Virus (SINV), Adenovirus Serotype 5 Vector (AdV5), Equine Arteritis Virus (EAV), and Measles Virus (MV) showed no significant effect, indicating the virus-specific effects of LY6E in enhancing infection [27]. Mechanistic studies further demonstrate that LY6E enhances the YFV infection by acting on early steps of the viral life cycle that are after viral attachment to the cell surface but before viral protein translation, replication, and production [27]. They were able to show that IAV uncoating is the key step to be targeted by LY6E [27]. 

The enhanced entry of flaviviruses by LY6E is consistent with a study showing that flavivirus internalization is facilitated by LY6E in osteosarcoma epithelial cell U2OS [28]. Interestingly, however, in the latter study, it was observed that the flavivirus infection induces LY6E tubularization of target cells, a process that resembles microtube assembly, and that uptake of large clathrin-dependent endocytosed cargoes is required [28]. In aligning with this finding, GPI-anchored proteins have been previously shown to be linked to cytoskeleton rearrangement [29]. Given that the engagement of cytoskeleton is involved and is sometimes critical for virus entry, it is likely that the GPI-anchored LY6E molecule can intersect, structurally or functionally, with the internalization process of viral particles; more experiments are needed to interrogate the likely complex process. 

In contrast to the positive role of LY6E in viral infection described above, LY6E has also been shown to inhibit viral infection (Table 1). For example, the replication of VSV, a negative-strand RNA virus, was shown to be restricted by LY6E in an ISG screening study [30]. In this work, it was shown that the overexpression of LY6E in HEK293 cells inhibits VSV replication by threefold yet has no effect on a single-round infection, suggesting that LY6E likely affects some late stages of VSV replication, such as viral protein trafficking/assembly and release. Interestingly, by using both single-round and replication-competent VSV, we found no significant effect of LY6E knockdown in A549 and T cells ([18]), arguing against an active role of LY6E in regulating VSV replication and entry. Noticeably, another recent study reported that the overexpression of LY6E in STAT1^-/-^ fibroblasts promotes VSV replication [27], again suggesting that the distinct phenotypes of LY6E reported by different groups may be due to specific cell types and/or the virus strains used in these experiments. A similar scenario could be true for murine gammaherpesvirus 68 (MHV-68), which has been shown to be modestly inhibited by LY6E overexpression [30].

## 5. Mechanisms of Action by LY6E on Viral Infection: Direct vs. Indirect Effects

A critical question is how LY6E modulates viral infection, and often in a cell type-dependent and virus-specific manner. We consider the following aspects that might be related to the general mechanisms of action by LY6E on viral infection. 

First, LY6E is a GPI-anchored protein, which is enriched in the lipid-raft microdomain of the plasma membrane. Similar to many other GPI-anchored proteins, which normally regulate membrane-dependent processes, such as endocytic trafficking and signaling [31], LY6E can directly or indirectly affect the expression, kinetics, or biophysical properties of cellular receptors for viruses or viral glycoproteins, thus effecting virus binding, trafficking, and membrane fusion. In the case of HIV-1, we have shown that LY6E promotes viral fusion and enhances entry by modulating viral hemifusion yet downregulates the cell surface receptor CD4 thus inhibiting the binding of HIV-1 to low CD4 cells and diminishing the viral entry. While HIV co-receptors have not been found to be affected by LY6E, molecules associated with CD4 and/or co-receptors could be influenced by LY6E.

Second, GPI-anchored LY6 proteins could modulate the cytoskeleton reengagement [32]. Although GPI-anchored proteins may, in theory, not directly interact with the cytoskeleton molecules, because of their inaccessibility to the cell interior, they can be associated with multiple transmembrane adaptors, such as Src family kinase [33] and integrin [34], which are present in the same microdomains or in a close proximity to LY6E, hence indirectly interacting and modulating the organization of cytoskeletons [35]. The enhancing or inhibitory effect of LY6E on infection by different viruses could be ascribed to indirect functions of LY6E in cytoskeleton rearrangements [27,28], the effect of which could be cell type-dependent and virus-specific. 

Third, LY6E can regulate cell signaling, including the host immune response, which is essential for defending against viral infections. It is well recognized that cytokines, chemokines, as well as their cognate receptors and associated adaptors are involved in type I IFN signaling, thus facilitating the clearance of invading pathogens through a series of signaling cascades [36]. In fact, LY6E signaling has been recognized as a modulator of T lymphocyte physiology for a long time. It was shown that antibody cross-linking of the cell surface LY6E activates T cells, and that this activation largely relies on the C-terminal GPI anchor [37]. Because replacement of the GPI anchor of LY6E with a transmembrane domain completely abolishes the crosslinking-induced T cell proliferation [37], it seems that the GPI anchor is involved in mediating the signaling event. However, it is debatable whether or not it is the GPI anchor that directly mediates the signal transduction, or maybe the GPI-anchor associates with a third molecule to fulfill this task. The latter notion is supported by the finding that T cell receptor (TCR/CD3) is required for the T cell activation by LY6E [38], and physical interaction between LY6E and the T cell receptor (TCR/CD3) ζ chains is important for this process [39]. Additional experiments showed that anti-LY6E antibody treatment leads to CD3ζ chain tyrosine phosphorylation [39], as well as blocks TCR-mediated T cell activation and apoptosis [6,7]. Notably, another GPI anchored protein CD48 has been reported to co-engage with CD3, leading to CD3ζ chain tyrosine phosphorylation and T cell activation [40]; this would suggest that the presence of GPI anchors, together with their associations with CD3, are important for the signaling induction. Given that T cell physiology, such as activation, proliferation and antigen recognition, can profoundly impact the onset and outcome of viral infection, and that the completion of productive infection by retroviruses, such as HIV, significantly depends on T cell activation [41], it would be interesting to explore whether LY6E signaling regulates the host adaptive immunity to viral infection by influencing the TCR-mediated antigen recognition in vivo. 

LY6E-associated cell signaling regulates viral infection. Yeom et al. reported that LY6E serves as a conductor of tumor growth through modulation of the PTEN/PI3K/Akt/HIF-1 axis [42], specifically by down-regulating PTEN yet up-regulating HIF-1α gene expression at the transcription level [42]. LY6E, together with LY6K, has also been implicated in breast cancer proliferation by altering TGFβ-dependent breast cancer cell physiology, resulting in increased immune checkpoint molecules PD-L1 and CTLA4 in tumor-infiltrating T regulatory cells yet decreased natural killer (NK) cell activation [43]. These signaling events can directly influence viral infection. Hepatitis C virus NS5A protein drives a PTEN/PI3K/Akt feedback loop to promote cell survival [44]. Epstein–Barr virus (EBV) can utilize its latent membrane protein 1 (LMP1) to trigger the PTEN/PI3K/Akt pathway and induce stem-like cells in nasopharyngeal carcinoma [45]. HSV-1 infection stimulates the PI3K/AKT signaling pathways, which in turn contributes to Kaposi's sarcoma-associated herpesvirus (KSHV) reactivation during the lytic cycle [46]. Because TGFβ is widely implicated in viral infection and pathogenesis [47], it would be important to explore the possible involvement of LY6E in TGFβ-regulated viral infections. 

A direct interaction between LY6E and innate immune signaling has been recently demonstrated. Specifically, LY6E was shown to down-regulate CD14, a key molecule involved in the TLR4/CD14/NF-κB pathway [17], and this results in a negative feedback loop in innate immune activation [17] and increased viral pathogenesis. While the experiment was performed in the study of HIV, the implication could be applied to other viral infections, as an adequate inflammatory response can be protective against virus infections. We recently found that LY6E promotes HIV-1 gene expression, likely by acting on the LTR promoter region of the viral genome [18]. While it remains to be determined as to how LY6E modulates the LTR activity, LY6E likely regulate HIV expression by altering cellular transcription factors. Given that NF-κB is known to be essential for the lentiviral transcription [48,49], it is reasonable to speculate that LY6E may affect HIV-1 gene expression by interfering with the NF-κB level in virus-infected cells. It should be noted that although LY6E is a plasma membrane-targeting protein, it is also expressed intracellularly, including on intracellular membranes as examined by immunofluorescence staining [24,28,50]. In this respect, it will be informative to explore how this portion of LY6E may physically interact with transcription factors, thus participating in the modulation of viral and cellular gene expression.

## 6. Concluding Remarks and Future Perspectives

The role of LY6E in viral infection has been studied for different viruses and the effect appears to be dependent on specific viruses, and in some cases specific cell types used for experiments. Nonetheless, evidence points to an important role of the GPI anchor of LY6E in modulating cellular receptors, viral proteins, cell signaling molecules, and endocytic trafficking. Variations in the abundance of expression, as well as the localization of LY6E and its associated proteins or lipids, could explain the different effects of LY6E on the infection of different viruses in different cells.

LY6E is one of the LY6/uPAR family members, and earlier studies have suggested that other members, in addition to LY6E, also contribute to modulating viral infection, including that of HIV-1 [16]. Phylogenetic analysis shows that LY6E is closely related to prostate stem cell antigen (PSCA) (Figure 2), which has been shown to modulate infection by YFV [27]—see an accompanying review in this issue. Thus, it will be important to examine the possible interplay between LY6E and other members of the Ly6/uPAR family in the context of viral infection, including the step of viral entry. While GPI anchors have been shown to be important for modulating infection by a large number of viruses, it is possible that the extracellular portion of LY6E may also critically regulate viral infection, either directly or indirectly, by associating with other molecules on the cell surface. Ultimately, the function and significance of LY6E in viral infection must be demonstrated *in vivo*. This is particularly important with regards to understanding the multifaced role of LY6E in regulating host innate and adaptive immunity to viral infection and viral pathogenesis.

## Figures and Tables

**Figure 1 viruses-11-01020-f001:**
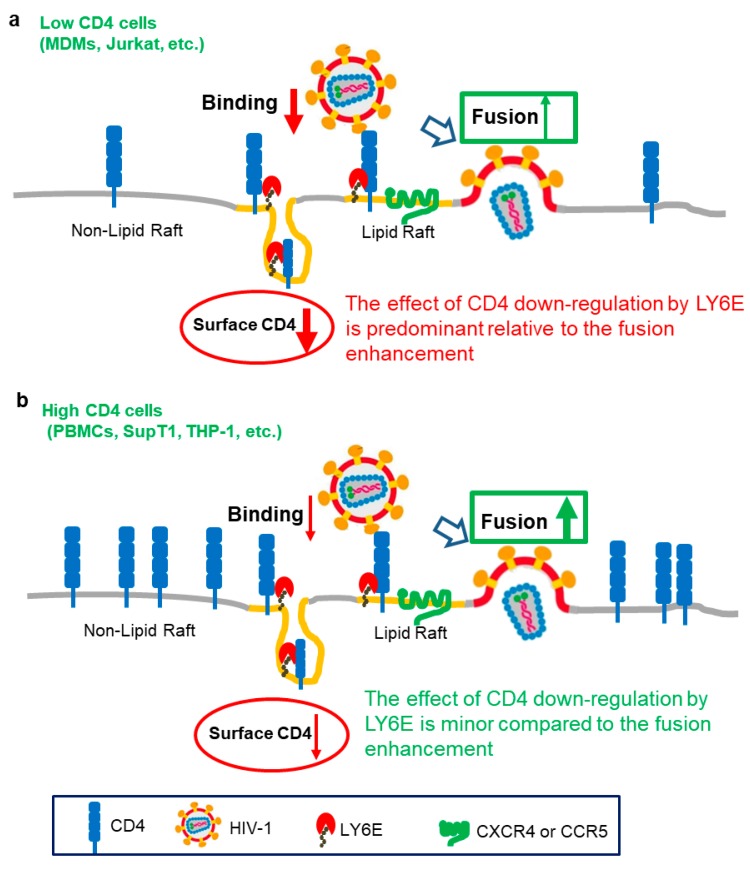
Working model for the differential effects of LY6E on HIV-1 infection. Adapted from Yu et al. [24]. (**a**) In low CD4-expressing cells (such as Jurkat T cells, macrophages and others), LY6E is associated with CD4 within the lipid-raft microdomain, thus promoting its internalization from the plasma membrane; this results in a decreased CD4 level on the cell surface, therefore impairing virus binding and entry. While LY6E still intrinsically promotes fusion in this case, the effect of LY6E on down-regulating CD4 is predominant, leading to an overall inhibition of HIV-1 infection. (**b**) In high CD4-expressing cells (such as PBMCs, SupT1 cells, CHME3 and others), the effect of LY6E on the down-regulating CD4 is minor compared to the LY6E-mediated enhancement of viral fusion. In this case, LY6E functions as a positive factor for HIV-1 infection. This latter mechanism may be related to the GPI-anchored topology of LY6E and its the raft localization.

**Figure 2 viruses-11-01020-f002:**
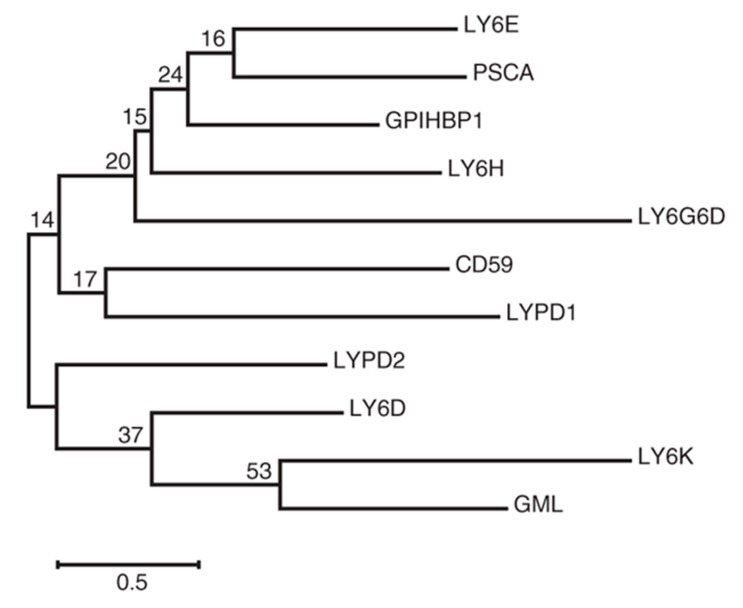
Phylogenetic analysis of human ly6/uPAR family proteins constructed by using the maximum likelihood method based on the Jones–Taylor–Thornton (JTT) matrix-based model. Adapted from reference [27].

**Table 1 viruses-11-01020-t001:** Summary of the effects of LY6E on viral infection.

Virus	Family	Mode of Action	Mechanism of Action	Tissue/Cell Type Tested	Reference
Mouse Adenovirus Type 1 (MAV-1)	Adenovirus	Enhanced	Enhance mouse susceptibility by genetic mapping	BALB/cJ Mice	[14,15]
Marek’s Disease Virus (MDV)	Herpesvirus	Enhanced	Enhance chicken susceptibility by genetic mapping	Chicken	[12]
Vesicular Stomatitis Virus (VSV)	Rhabdovirus	Restricted	Unknown	HEK293	[30]
		Enhanced	Unknown	STAT1^-/-^ fibroblasts, THP-1, U2OS	[27]
Zika Virus (ZIKV)	Flavivirus	Enhanced	LY6E tubularization facilitates the uptake of large clathrin-dependent endocytosed cargoes	U2OS, STAT1^-/-^ fibroblasts	[27,28]
Dengue Virus (DENV)	Flavivirus	Enhanced	LY6E tubularization facilitates the uptake of large clathrin-dependent endocytosed cargoes	U2OS, STAT1^-/-^ fibroblasts	[27,28]
Yellow Fever Virus (YFV)	Flavivirus	Enhanced	Enhancing an early stage of life cycle that is after attachment but before viral translation	STAT1^-/-^ fibroblasts, THP-1, U2OS	[27]
West Nile Virus (WNV)	Flavivirus	Enhanced	LY6E tubularization facilitates the uptake of large clathrin-dependent endocytosed cargoes	U2OS	[28]
Human Immunodeficiency Virus (HIV-1)	Lentivirus	Enhanced	Enhance viral entry, possibly acting on virus–cell membrane fusion	CD4 high T cells and PBMCs	[18]
		Restricted	Restricting HIV-1 infection by lowing the cell surface CD4	CD4 low Macrophages	[24]
Endogenous Retroviral Envelope, Syncytin-A	Retrovirus	Enhanced	Facilitating cell–cell fusion by serving as the syncytin-A receptor	Murine syncytiotrophoblast	[22]
Influenza A Virus (IAV)	Orthomyxovirus	Enhanced	Enhancing uncoating	U2OS	[27]
O'nyong'nyong Virus (ONNV)	Alphavirus	Resistant	Unknown	STAT1^-/-^ fibroblasts	[27]
Sindbis Virus (SINV)	Alphavirus	Resistant	Unknown	STAT1^-/-^ fibroblasts	[27]
Equine Arteritis Virus (EAV)	Alphaarterivirus	Resistant	Unknown	STAT1^-/-^ fibroblasts	[27]
Measles Virus (MV)	Paramyxovirus	Resistant	Unknown	STAT1^-/-^ fibroblasts	[27]
Parainfluenza Virus-5 (PIV5)	Paramyxovirus	Resistant	Unknown	U2OS	[28]
Replication-Defective Adenovirus Serotype 5 Vector (AdV5)	Adenovirus	Resistant	Unknown	STAT1^-/-^ fibroblasts	[27]

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
