# Peer review of "Emerging Role of LY6E in Virus–Host Interactions"

_viruses, 2019, doi:10.3390/v11111020_

Round 1
Reviewer 1 Report
The review written by Jingyou Yu and Shan-Lu Liu summarized the role of LY6E in multiple virus infections. This includes the opposing role of LY6E in HIV entry process, modulation of other virus infections by LY6E, and several possible mechanisms of action by which LY6E regulates virus infections. The review is well organized, interesting, and informative, compiling diverse roles of LY6E during viral infections. A few minor suggestions are made below.
Lines 206 and 210: “antigen presentation” should change to “antigen recognition (or antigen engagement or binding)”, since T cells are not traditional antigen presenting cells.
Line 229: it is difficult to agree with the sentence “inflammatory response is generally regarded as a protective mechanism against viral infection.” As the authors are aware, the inflammatory response can be often pathogenic to the host during virus infections. Better to rephrase it, e.g., an adequate inflammatory response can be protective against virus infections; early inflammatory response could activate antiviral innate immunity against virus infections, etc.
Line 104, “one the other” to “on the other”; Lines 125 and 132, STAT-/- to STAT1-/-; Line 156, delete “has”; Line 216, ye to yet; Line 253, shown to be important.
Author Response
We thank the reviewer for his/her positive and helpful comments. See below our response point by point.
Lines 206 and 210: “antigen presentation” should change to “antigen recognition (or antigen engagement or binding)”, since T cells are not traditional antigen presenting cells.
Response: We agree and have changed “antigen presentation” to “antigen recognition” as suggested.
Line 229: it is difficult to agree with the sentence “inflammatory response is generally regarded as a protective mechanism against viral infection.” As the authors are aware, the inflammatory response can be often pathogenic to the host during virus infections. Better to rephrase it, e.g., an adequate inflammatory response can be protective against virus infections; early inflammatory response could activate antiviral innate immunity against virus infections, etc.
Response: We absolutely agree with the reviewer’s point and have changed the sentence as follows: “inflammatory response is generally regarded as a protective mechanism against viral infection” to “an adequate inflammatory response can be protective against virus infections”.
Line 104, “one the other” to “on the other”; Lines 125 and 132, STAT-/- to STAT1-/-; Line 156, delete “has”; Line 216, ye to yet; Line 253, shown to be important.
Response: We have corrected all these typos as suggested. Thank you.
Reviewer 2 Report
Interesting, well-written paper that discusses the role of LY6E in host-virus interaction.
No major/minor criticisms regarding the paper.
On line 253.. “…been shown to be important…” insert be…
Author Response
We thank the reviewer for his/her positive and helpful comments. See below our response point by point.
On line 253.. “…been shown to be important…” insert be…
Response: We have corrected the typo as suggested. Thank you.